# Student and Tutor Satisfaction with Problem-Based Learning in Azerbaijan

**Ulkar Sattarova** [1,2,*] **, Wim Groot** [1] **and Jelena Arsenijevic** [3]

1   School of Governance, Maastricht University, 6200 Maastricht, The Netherlands;
    wim.groot@maastrichtuniversity.nl
2   Department of Information Technologies and Systems, Azerbaijan University of Architecture and
    Construction, Baku 1043, Azerbaijan
3   School of Governance, Utrecht University, 3508 Utrecht, The Netherlands; j.arsenijevic@uu.nl
*   Correspondence: u.sattarova@student.maastrichtuniversity.nl

**Abstract:** This article examines tutors' and students' satisfaction with the implementation of problem-based learning (PBL) at the Azerbaijan University of Architecture and Construction. A total of 28 pilot academic staff members and their students participated in PBL during one semester and received a questionnaire about their experiences at the end of the semester. In total, 649 students were involved in the intervention. Descriptive statistics and factor analyses were used to analyze the data. In total, the response rate among students was 61.7%, and 82.1% among tutors. More than 83% of the students thought that the PBL should be kept as part of the module. A total of 91.3% of tutors agreed that PBL is a great tool for student learning. According to the factor analysis, tutors believed that PBL can develop students' ability for group/team work. Tutors also identified some barriers in applying PBL. For example, they mentioned a lack of relevant skills to apply PBL in higher education. Both students and tutors found the PBL to be a suitable learning tool for their curriculum.

**Keywords:** higher education; Azerbaijan; problem-based learning; PBL; student and tutor satisfaction





## 1. Introduction

PBL can help to change teaching from simply reproducing what was learned to critical thinking and self-development. During the 1970s, PBL was viewed as a pedagogical innovation [1]. Three variants of PBL are distinguished in the literature: the Canadian (McMaster), the Dutch (Maastricht), and the Danish approaches [2]. The PBL-based curriculum was first launched at McMaster University in Hamilton in 1969. It attracted international attention, especially from the newly established university in Maastricht in the Netherlands. In parallel to the development of PBL at McMaster and Maastricht, a version of PBL was developed in Denmark. The PBL approach in Denmark was specific since it was the first PBL application in engineering education at the University of Alborg in Denmark. In the Danish and Maastricht models, students first analyze and define the problem within a given domain or interdisciplinary context. A focus on the community is typical in the Danish model. Although there are some minor differences between the three universities, they all emphasize three common elements: group work (teamwork), problem-oriented learning, and a community-oriented attitude. The Danish approach differs because of its emphasis on an interdisciplinary approach [3]. It means that the problem (task) is not analyzed from the view of one disciple (subject) only. In the Danish and Maastricht models, students first analyze and define the problem within a given domain or interdisciplinary context. For example, in the Danish model (and in some studies at Maastricht University also), the degree of self-direction of students is higher, and there is a greater focus on skills (e.g., planning, monitoring) and the application of knowledge to real-life situations. In general, PBL supports students to obtain academic knowledge through real life cases. At the same time, it introduces them to research,

on how to use and understand research findings, but also on how to understand and use data [4].

The Azerbaijan University of Architecture and Construction is one of the leading state universities in Azerbaijan. It started the Bologna Process in 2005 [5]. Before that, education at the university was conducted according to the traditional Soviet teaching system. This system, inherited from the ex-Soviet Union, had several typical features such as an emphasis on academic lectures in auditorium, up to 10 non-core subjects, significantly more hours of lectures than practical/laboratory/project lessons, the absence of internship programs, the absence of lecturers with a practical background, a short period of time before exams, lack of departments/centers for career planning, an education process that is immune from developments in business and economics, and the absence of a system of credit points [5]. One of the main problems was a huge gap between theoretical knowledge and real practical competencies and skills, which are necessary for occupation and career development. A typical feature was the strong theoretical emphasis and little attention being paid to practical application and to the acquisition of skills. A major reason for this was the lack of laboratories and techno centers for teaching practical skills. Furthermore, there was an absence of university–industry collaboration and a lack of interest in businesses and entrepreneurial activities among university faculty and management. Although universities in the former Soviet states have recognized the need for change in the educational process, the implementation of the Bologna Process was not sufficiently successful as most decisions to change the system faced huge barriers in implementation and were opposed and blocked by educational traditions [5].

The higher education system in Azerbaijan does not seem to be able to prepare students well for the labor market. One of the main indicators demonstrating this is the high unemployment rate of university graduates. The number of unemployed people who have completed secondary education is almost equal to the number of unemployed people with a higher education degree [6]. Thus, a university level education and a recognized diploma do not provide many benefits. Higher education appears not to provide the necessary skills for the labor market or to prepare graduates for start-up firms and entrepreneurial activities [7]. As indicated above, there is a necessity to change and modernize the content and format of the educational process in Azerbaijan. One way of doing this is by applying more modern educational approaches such as PBL [8,9].

Moreover, the level of satisfaction with the current higher education system is low. Therefore, it was decided that the PBL approach would be introduced in "pilot" groups of students. A pilot group of tutors agreed to support this. PBL provides students not only with theoretical knowledge but also with practical skills. It is built on an education process that is more democratic than the classic Soviet traditional way of education [10,11]. PBL helps students to develop soft skills by activating learning and discussing problems and working together in tutorial groups.

There are some difficulties in applying PBL in post-Soviet countries, as for example the fact that university–industry collaboration is not part of the university culture. This limits the ability of tutors to develop real-life cases based on the state of the art in the industry. Furthermore, reporting findings from the literature and an analytical approach to the literature are not part of the higher education curriculum. The number of hours spent on a subject/module depends on available staff. Teaching is mainly based on lectures, which only consist of lecturers explaining theory. Practically oriented lessons based on laboratory experiments, project-based lessons with research and analytical work, using real problems and cases, are only a minor part of the curriculum. Interactive lessons are not accepted as traditional lectures where the lecturer addresses the students. The average age of lecturers is around 65 years. This poses difficulties in the application of more modern teaching methods, as these frequently require change, adaptation, and special training, as well as a change in culture for tutors and lecturers.

To the best of our knowledge, there are no studies on the application of PBL in countries such as Azerbaijan that so heavily depend on traditional forms of teaching in

higher education. To modernize higher education in Azerbaijan, we chose a selected group of academic staff members of AzUAC (14.02.2.19, www.azmiu.edu.az, accessed on 4 April 2020) to participate in a pilot application of PBL during the second semester of the 2020 academic year. The group was selected according to the interest they had shown in innovation in teaching in their previous work. Most of them were already involved in projects supported by Europe (such as Erasmus, Tempus, Backis), government financed education/scientific grants, or university internal activities/projects. They already had experience with different didactic approaches and motivation to participate in this pilot application of PBL.

First, a paper-based survey about the current knowledge of tutors was conducted to understand the current situation. In this paper, we do not report results from this initial survey. Second, during an info day, the pilot tutors of AzUAC were familiarized with the main concept of PBL in teaching by presentations based on Maastricht University (www.maastrichtuniversity.nl/, accessed on 4 April 2020) materials about PBL. At the same time, incentives such as financial support for personal development were suggested for the best tutors in the group. This was done to motivate staff members. Third, the next winter semester was selected as a probation period where the PBL intervention was applied. After the info day, three two-day-long special workshops were organized for tutors during the winter semester to support the application of PBL. During these events and sessions, practical tips and cases were provided and discussed [12].

The PBL group of academic staff members, consisting of 26 tutors, was formalized. A WhatsApp group was created to share experiences among the tutors during the experimental semester and to encourage each other. Therein, tutors were enabled to make short videos from their classes and shared these videos (students were informed and appropriate permission was received). At the same time, it helped to better manage and share information. After the pilot PBL semester, tutors filled in a questionnaire about their experiences with PBL.

Furthermore, at the beginning of the winter semester, students also received information about PBL. Tutors started to teach their courses by the PBL method. At the end of the course, students also received questionnaires about their experiences with PBL.

Because of the COVID-19 pandemic, lessons and some of the training sessions were conducted by Zoom, and because of some technical issues with distance learning, a few days were wasted. Furthermore, the pandemic and the complete lockdown influenced individuals and presented additional difficulties and opportunities in teaching.

The aim of this paper was to examine and to report on the experiences with PBL of tutors' and students' communities at Azerbaijan University of Architecture and Construction. For this purpose, we used the data from surveys collected in the spring semester of the 2019/2020 academic year (from February to May).

In this paper, we focus on satisfaction with PBL among students and tutors. The reason for doing so is that satisfaction is a prerequisite for an effective implementation of PBL [13]. A reform of the traditional curriculum is only possible if students and tutors support and are satisfied by the alternative PBL method. There have been a number of previous studies on the satisfaction with PBL among students and tutors. Student satisfaction with PBL is evidently positive [13]. Previous studies have found that PBL has positive effects on student performance and interest, exam scores, effort, the ability of working in groups, improvements in soft skills, and student satisfaction [13,14]. Athene complexity of the use of PBL in traditional teaching contexts has also been previously documented in the literature [7,15].

## 2. Methods

After the pilot semester with PBL, the surveys about students' and tutors' satisfaction were conducted. As PBL was newly introduced, it is relevant to know how satisfied students and tutors are with this, for them, new teaching method. Furthermore, the wider implementation of new teaching methods such as PBL is likely to be more successful and more supported if students and tutors are satisfied with it.

The surveys, on which findings we report here, were conducted in the last weeks of the semester, i.e., in May and June, among students and tutors. The design of the PBL intervention in AzUAC is described in Figures A1 and A2 in the Appendix A. descriptive statistics such as means, standard deviations, and minimum and maximum values for each question were calculated. The descriptive table were performed with the statistical software SPSS.

*Student survey.* In order to measure student satisfaction, we included in the questionnaire several questions on the satisfaction with the organization of the subject, the instructiveness of the subject, the link of the subject with your prior knowledge, the productivity of the tutorial group, the link between the subject and the assessment, the quality of tasks/problems/cases, and the quality of lectures. In total, the questionnaire consisted of 26 questions (see Supplementary Materials S#3). Similar questions were applied in the previous research (ref.). In total, the response rate was 61.7%, with 401 completed questionnaires. The survey was conducted in electronic form only. We were also interested in students' ideas and personal comments. Therefore, we included some open questions as well.

To analyze the data, Descriptive statistics were calculated, factor analysis was performed, and Cronbach's alphas were calculated to assess the reliability of the factors.

*Tutor survey.* The questionnaire for tutors was based upon an existing questionnaire [16] (Supplementary Materials S#4). In total, the survey consisted of 65 questions. We carried out a pilot survey among 5 randomly selected tutors before presenting the survey to all tutors. There was a pilot group of tutors who agreed to fill in the main questionnaire and to give their comments to improve the survey. Some corrections in the questionnaire were made according to the valuable comments of the tutors. By their suggestion, the survey was prepared in Google Forms. This enabled us share the link easily in the WhatsApp group and to export the data to SPSS.

We included questions on the satisfaction with the pedagogical method, supervising problem processing in tutorial groups, potential barriers to student learning in PBL, the tutor's role in the tutorial group, and the relationship between theory and practice in PBL.

We also included questions on socio-demographic characteristics such as gender, education, and number of years worked in academia. The response rate was 82.1%, with 23 completed questionnaires from tutors. The descriptive statistics consisted of frequency distributions and other descriptive characteristics for each question (Table 4). To measure the internal consistency, we applied factor analysis and the Cronbach's alpha (Tables 5 and 6). An open question was also included in the survey. We asked tutors 'What additional problems have you had?'

## 3. Results

### 3.1. Student Survey

Below are the results of the descriptive statistics of the students' survey with the frequency distribution for each question. No one indicated that the organization or instructiveness was very poor. Students reported that PBL helped them to make a link between practical cases and theoretical materials. Moreover, more than 84% of respondents agreed that it supported them in group working.

Students in general said that PBL helped them to formulate clear learning goals themselves. Around 77% of students who participated in the intervention said that PBL lessons were well organized and effective. More than 83% of students who participated in the intervention said that PBL should be part of this subject/module. Almost 73% of students mentioned that this approach was helpful in understanding lectures. However, there were 11% of students think thought that tutors did not motivate them to summarize what they learned in their own words enough. Moreover, almost 9% of students believed that tutors (very) poorly helped them to apply what they had learned to the task, situation, or problem. As is evident from Table 1, the mean was around four for almost all questions, indicating that, on average, the 'good' answer was selected by respondents. The standard deviation was low and between 0.64–1.16, indicating that most responses were close to the average.

**Table 1.** Descriptive statistics for items related to students' opinion (*n* = 401).

| Questions | Frequency Distribution % (*n*) | | | | | Statistical Characteristics | | | |
| | **1** | **2** | **3** | **4** | **5** | | | | |
| | *Very Poor* | *Poor* | *Normal* | *Good* | *Excellent* | | | | |
| | *Too Easy* | *Easy* | *Just Right* | *Difficult* | *Too Difficult* | **M** | **SD** | **Min** | **Max** |
| | **Fully Disagree** | **Disagree** | **I Don't Know** | **Agree** | **Fully Agree** | | | | |
| The organization of the subject | - | 1.2 (5) | 4.5 (18) | 36.7 (147) | 57.6 (231) | 4.51 | 0.64 | 2 | 5 |
| The instructiveness of the subject | - | 7 (28) | 4.7 (19) | 36.9 (148) | 51.4 (206) | 4.33 | 0.86 | 2 | 5 |
| The link of the subject with your prior knowledge | 1.0 (4) | 3.2 (13) | 19.2 (77) | 37.7 (151) | 38.9 (156) | 4.10 | 0.89 | 1 | 5 |
| The productivity of the tutorial group | 0.5 (2) | 1.0 (4) | 11.0 (44) | 32.9 (132) | 54.6 (219) | 4.40 | 0.76 | 1 | 5 |
| The link between the subject and the assessment | 4.0 (16) | 2.2 (9) | 6.7 (27) | 32.9 (132) | 54.1 (217) | 4.31 | 0.98 | 1 | 5 |
| The quality of tasks/problems/cases | 1.5 (6) | 1.2 (5) | 9.5 (38) | 34.2 (137) | 53.6 (215) | 4.37 | 0.82 | 1 | 5 |
| The quality of lectures | 1.5 (6) | 3.7 (15) | 8.7 (35) | 32.4 (130) | 53.6 (215) | 4.33 | 0.90 | 1 | 5 |
| The subject contents were? | 3.2 (13) | 14.0 (56) | 38.7 (155) | 35.9 (144) | 8.2 (33) | 3.32 | 0.93 | 1 | 5 |
| How many hours on average did you spend this subject on self- study per week? | - | - | - | - | - | 2.85 | 1.59 | 1 | 8 |
| How many hours on average did you spend this subject on your study in total per week | - | - | - | - | - | 2.78 | 1.76 | 1 | 8 |
| The tutor stimulated us to summarize what we had learned in our own words | 1.7 (7) | 9.2 (37) | 23.2 (93) | 48.1 (193) | 17.7 (71) | 3.71 | 0.92 | 1 | 5 |
| The tutor stimulated us to create links between the contents of the different parts of the subject matter | 1.7 (7) | 3.7 (15) | 18.5 (74) | 42.1 (169) | 33.9 (136) | 4.03 | 0.91 | 1 | 5 |
| The tutor stimulated us to formulate clear learning goals ourselves | 0.7 (3) | 3.5 (14) | 14.7 (59) | 46.1 (185) | 34.9 (140) | 4.11 | 0.83 | 1 | 5 |
| The tutor stimulated us to apply what we had learned to the task/other situations/problems | 1.5 (6) | 7.0 (28) | 16.2 (65) | 40.6 (163) | 34.7 (139) | 4.00 | 0.96 | 1 | 5 |
| The tutor stimulated us to provide constructive feedback during the tutorial meetings | 1.2 (5) | 2.5 (10) | 23.4 (94) | 37.2 (149) | 35.7 (143) | 4.03 | 0.90 | 1 | 5 |

**Table 1.** *Cont.*

| Questions | Frequency Distribution % (*n*) | | | | | Statistical Characteristics | | | |
|---|---|---|---|---|---|---|---|---|---|
| | *1* | *2* | *3* | *4* | *5* | | | | |
| | *Very Poor* | *Poor* | *Normal* | *Good* | *Excellent* | | | | |
| | *Too Easy* | *Easy* | *Just Right* | *Difficult* | *Too Difficult* | *M* | *SD* | *Min* | *Max* |
| | **Fully Disagree** | **Disagree** | **I Don't Know** | **Agree** | **Fully Agree** | | | | |
| The tutor stimulated us to regularly evaluate the way we cooperated in the tutorial group | 1.5 (6) | 4.2 (17) | 18.5 (74) | 42.4 (170) | 33.4 (134) | 4.02 | 0.91 | 1 | 5 |
| The PBL sessions have improved my understanding of the lectures provided within this module | 1.5 (6) | 9.5 (38) | 17.0 (68) | 43.6 (175) | 28.4 (114) | 3.88 | 0.98 | 1 | 5 |
| The PBL sessions have helped my understanding of the theoretical network design process | 1.2 (5) | 5.5 (22) | 19.0 (76) | 49.6 (199) | 24.7 (99) | 3.91 | 0.87 | 1 | 5 |
| The PBL sessions have improved my understanding of the practical aspects of network design | 3.2 (13) | 6.5 (26) | 22.7 (91) | 42.9 (172) | 24.7 (99) | 3.79 | 0.99 | 1 | 5 |
| Having participated in the PBL sessions, my confidence and ability to undertake a real network design has been enhanced | 4.5 (18) | 5.0 (20) | 20.7 (83) | 46.4 (186) | 23.4 (94) | 3.79 | 1.00 | 1 | 5 |
| The PBL sessions were realistic and reflected typical real practical situations | 3.7 (15) | 4.7 (19) | 18.7 (75) | 41.4 (166) | 31.4 (126) | 3.92 | 1.01 | 1 | 5 |
| The PBL sessions have helped my ability to work in groups | 6.0 (24) | 10.0 (40) | 20.7 (83) | 35.2 (141) | 28.2 (113) | 3.70 | 1.16 | 1 | 5 |
| The PBL sessions were well organized and effective | 1.2 (5) | 3.7 (15) | 18.5 (74) | 43.6 (175) | 32.9 (132) | 4.03 | 0.88 | 1 | 5 |
| The PBL sessions should be kept as part of this module | 1.7 (7) | 3.0 (12) | 12.2 (49) | 43.4 (174) | 39.7 (159) | 4.16 | 0.88 | 1 | 5 |

Reliability statistics are shown in Table 2. The Cronbach's alpha was above 0.9, showing high internal consistency and a good reliability of the items included.

**Table 2.** Reliability statistics. Cronbach's alpha result of students' survey.

| Cronbach's Alpha | Cronbach's Alpha Based on Standardized Items | *n* of Items |
|---|---|---|
| 0.913 | 0.914 | 22 |

The factor analysis in SPSS suggested initially that there were five underlying factors. However as three of them only had only one or two items that belonged to these factors, we decided for better interpretation to limit it to two factors. The results of the factor analysis of students' survey are shown below. Table 3 describes the factor analysis of students' main survey. The variance and component transformation matrix are also presented (Appendix A, Tables A1 and A2).

**Table 3.** Factor analysis of students' survey.

| Question #in Questionnaire | Question | Factors | | | |
|---|---|---|---|---|---|
| | | Factor 1 | Factor 2 | Factor 1 | Factor 2 |
| Q_1 | The organization of the subject | 0.1 | **0.744** | | ✓ |
| Q_2 | The instructiveness of the subject | 0.103 | **0.784** | | ✓ |
| Q_3 | The link of the subject with your prior knowledge | 0.053 | **0.573** | | ✓ |
| Q_4 | The productivity of the tutorial group | 0.245 | **0.552** | | ✓ |
| Q_5 | The link between the subject and the assessment | 0.095 | **0.768** | | ✓ |
| Q_6 | The quality of tasks/problems/cases | 0.255 | **0.482** | | ✓ |
| Q_7 | The quality of lectures | 0.136 | **0.691** | | ✓ |
| Q_8 | The subject contents were? | −0.009 | **−0.450** | | ✓ |
| Q_11 | The tutor stimulated us to summarize what we had learned in our own words | 0.189 | **0.662** | | ✓ |
| Q_12 | The tutor stimulated us to create links between the contents of the different parts of the subject matter | 0.399 | **0.626** | | ✓ |
| Q_13 | The tutor stimulated us to formulate clear learning goals ourselves | 0.441 | **0.591** | | ✓ |
| Q_14 | The tutor stimulated us to apply what we had learned to the task/other situations/problems | 0.368 | **0.678** | | ✓ |
| Q_15 | The tutor stimulated us to provide constructive feedback during the tutorial meetings | 0.493 | **0.522** | | ✓ |
| Q_16 | The tutor stimulated us to regularly evaluate the way we cooperated in the tutorial group | **0.567** | 0.41 | ✓ | |
| Q_17 | The PBL sessions have improved my understanding of the lectures provided within this module | **0.716** | 0.355 | ✓ | |
| Q_18 | The PBL sessions have helped my understanding of the theoretical network design process | **0.821** | 0.13 | ✓ | |
| Q_19 | The PBL sessions have improved my understanding of the practical aspects of network design | **0.794** | 0.208 | ✓ | |
| Q_20 | Having participated in the PBL sessions, my confidence and ability to undertake a real network design has been enhanced | **0.798** | 0.126 | ✓ | |
| Q_21 | The PBL sessions were realistic and reflected typical real practical situations | **0.773** | 0.054 | ✓ | |
| Q_22 | The PBL sessions have helped my ability to work in groups | **0.648** | 0.166 | ✓ | |
| Q_23 | The PBL sessions were well organized and effective | **0.748** | 0.264 | ✓ | |
| Q_25 | The PBL sessions should be kept as part of this module | **0.737** | 0.031 | ✓ | |

Extraction method: principal component analysis.
Rotation method: Varimax with Kaiser Normalization.

a. Rotation converged in 3 iterations.

Factor 1 included questions Q16, Q17, Q18, Q19, Q20, Q21, Q22, Q23, Q25, while factor 2 included Q1, Q2, Q3, Q4, Q5, Q6, Q7, Q8, Q11, Q12, Q13, and Q15. Factor 1 refers to the quality of teaching and shows that students in general had a positive attitude towards PBL, while factor 2 refers to the learning outcomes. Thus, to the questions Q11 and Q12 on comprehension outcome, 66% and 76% of the students answered agree/fully agree, respectively. As for Q13, which supports their ability for evaluation, more than 81% of respondents thought that the tutor stimulated this. As for the questions, Q14 and Q15 showed that more than 75% of students agreed/fully agreed on the positive realization from the tutors' side as well. Thus, students evaluated learning outcomes in the 'PBL semester' as quite positive.

In the open question (Supplementary Materials S#3 question 11) 'Tips to improve this subject', students mentioned some barriers in their learning such as weak library facilities, absence of the possibility to see videos in the auditorium, lack of practical and case-related tasks in the syllabuses, lack of tradition to invite guest speakers from business or industry sectors, and issues in communication with the tutor because of the pandemic. As for the second open question, 'Tips to improve this subject for the tutor' (Supplementary Materials S#3 question 26), students advised lectures be more oriented to real problems. At the same time, they advised making all lessons (not only experimental ones) more interactive, providing more possibilities to work in tutorial groups, trying to make lessons fun and not boring, applying a more interactive way of teaching, and improving the means of explanation in distance mode.

### 3.2. Tutor Survey

Below are the results of the descriptive statistics of the staff survey. As is evident from the descriptive table, academic staff members agreed that PBL as a didactic approach is very useful. With regards to the supervising problem processing by tutors, the opinions were positive too. According to the tutors' opinion, PBL is based on real-life cases and helps to develop practical skills. The link between lectures and practically oriented lessons is more evident with PBL. Although all tutors agree that group discussion facilitates problem processing and helps students share experiences with each other, three tutors said that this might be stressful for students.

Tutors also identified some barriers in applying PBL. For example, they mentioned a lack of relevant skills to apply PBL in higher education. Moreover, they were motivated to obtain more detailed training in order to acquire more knowledge about PBL. Furthermore, 19 respondents out of 23 agreed that they need more time and resources in order to become familiar with PBL. They also expressed the need for more specific materials for teaching PBL. No one fully disagreed about this matter, and 19 agreed or fully agreed. Moreover, 20 tutors said they were interested in training, and only 2 were doubtful about this. All 23 tutors had plans to apply PBL in their future teaching. Furthermore, there were five individuals that doubted if they were really well prepared for the PBL semester. The majority of the tutors were sure that this method has many advantages compared with the traditional way of teaching. As is evident from Table 4, the mean was around three to four almost in all questions, indicating that on average the 'good' was selected among respondents. The standard deviation was low, indicating that most answers were close to the average.

**Table 4.** Descriptive statistics for items related to tutors' second opinion survey (*n* = 23).

| Questions | Frequency Distribution % (n) | | | | | Statistical Characteristics | | | |
|---|---|---|---|---|---|---|---|---|---|
| **1—Fully Disagree to 5—Fully Agree** | | | | | | | | | |
| **PBL as Pedagogical Method** | **1** | **2** | **3** | **4** | **5** | **M** | **SD** | **Min** | **Max** |
| | **Fully Disagree** | **Disagree** | **I Don't Know** | **Agree** | **Fully Agree** | | | | |
| PBL helps the student acquire relevant knowledge for their profession | - | - | - | 60.9 (14) | 39.1 (9) | 4.39 | 0.5 | 4 | 5 |
| PBL contributes to the independence of students | - | - | - | 52.2 (12) | 47.8 (11) | 4.48 | 0.51 | 4 | 5 |
| Group tutorials help students to evaluate their own knowledge | - | - | 17.4 (4) | 52.2 (12) | 30.4 (7) | 4.13 | 0.69 | 3 | 5 |
| Group tutorials enrich student learning (communication and reflection) | - | 4.3 (1) | 4.3 (1) | 47.8 (11) | 43.5 (10) | 4.3 | 0.76 | 2 | 5 |
| Group tutorials help students share experiences with each other | - | - | - | 34.8 (8) | 65.2 (15) | 4.65 | 0.49 | 4 | 5 |
| In group tutorials, the students have time to sort out issues that are hard to understand | - | 4.3 (1) | 4.3 (1) | 34.8 (8) | 56.5 (13) | 4.43 | 0.79 | 2 | 5 |
| Group discussions help problem processing | - | - | - | 65.2 (15) | 34.8 (8) | 4.35 | 0.49 | 4 | 5 |
| Work in tutorial group helps students to reach an optimal depth of knowledge | - | 4.3 (1) | 21.7 (5) | 39.1 (9) | 34.8 (8) | 4.04 | 0.88 | 2 | 5 |
| In my opinion, PBL is a great tool for student learning | - | - | 8.7 (2) | 30.4 (7) | 60.9 (14) | 4.52 | 0.67 | 3 | 5 |
| *Supervising problem processing in tutorial groups* | | | | | | | | | |
| I support student learning by helping them to achieve the learning goals | - | - | - | 60.9 (14) | 39.1 (9) | 4.39 | 0.5 | 4 | 5 |
| I help the students to fulfill the aims of the course | - | - | - | 43.5 (10) | 56.5 (13) | 4.57 | 0.51 | 4 | 5 |
| I function as a resource person in the group | - | 4.3 (1) | - | 34.8(8) | 60.9 (14) | 4.13 | 0.97 | 2 | 5 |
| I participate in creating a positive work environment for the group | - | 4.3 (1) | - | 30.4 (7) | 65.2 (15) | 4.57 | 0.73 | 2 | 5 |
| I encourage student learning by stimulating questions | - | 4.3 (1) | - | 34.8 (8) | 60.9 (14) | 4.52 | 0.73 | 2 | 5 |
| I stress the importance of constant student reflection | - | 4.3 (1) | 13.0 (3) | 43.5 (10) | 39.1 (9) | 4.17 | 0.83 | 2 | 5 |
| I see to it that all students in the group have their say | - | 17.4 (4) | 13.0 (3) | 17.4 (4) | 52.2 (12) | 4.04 | 1.19 | 2 | 5 |
| I am sensitive to the wishes of the students regarding their need for support | - | - | 4.3 (1) | 52.2 (12) | 43.5 (10) | 4.39 | 0.58 | 3 | 5 |
| I am interested in being a tutor | - | - | 13.0 (3) | 26.1 (6) | 60.9 (14) | 4.48 | 0.73 | 3 | 5 |
| *Potential barriers to student learning in PBL* | | | | | | | | | |
| I have relevant teaching qualifications in PBL | - | 13.0 (3) | 17.4 (4) | 60.9 (14) | 8.7 (2) | 3.65 | 0.83 | 2 | 5 |
| It is difficult for students to know if they have learned enough | - | 52.2 (12) | 30.4 (7) | 17.4 (4) | - | 2.65 | 0.78 | 2 | 4 |
| Discussions in the tutorial group are slow-moving | 17.4 (4) | 60.9 (14) | 8.7 (2) | 4.3 (1) | 8.7 (2) | 2.26 | 1.1 | 1 | 5 |
| Work in the tutorial group has a test function and is stressful for students | 17.4 (4) | 60.9 (14) | 8.7 (2) | 13.0 (3) | - | 2.17 | 0.89 | 1 | 4 |
| Time for discussion in the tutorial group is too short | 4.3 (1) | 47.8 (11) | 17.4 (4) | 17.4 (4) | 13.0 (3) | 2.87 | 1.18 | 1 | 5 |

**Table 4.** *Cont.*

| | 1—Fully Disagree to 5—Fully Agree | | | | | | | | |
|---|---|---|---|---|---|---|---|---|---|
| **Questions** | **Frequency Distribution % (n)** | | | | | **Statistical Characteristics** | | | |
| **PBL as Pedagogical Method** | **1** | **2** | **3** | **4** | **5** | **M** | **SD** | **Min** | **Max** |
| | **Fully Disagree** | **Disagree** | **I Don't Know** | **Agree** | **Fully Agree** | | | | |
| The group size is just right from a tutorial point of view | 4.3 (1) | 26.1 (6) | 13.0 (3) | 47.8 (11) | 8.7 (2) | 3.3 | 1.11 | 1 | 5 |
| Discussion in the tutorial group creates uncertainty among students | 8.7 (2) | 52.2 (12) | 13.0 (3) | 21.7 (5) | 4.3 (1) | 2.61 | 1.08 | 1 | 5 |
| PBL evokes feelings of inadequacy in students | 8.7 (2) | 56.5 (13) | 21.7 (5) | 8.7 (2) | 4.3 (1) | 2.43 | 0.95 | 1 | 5 |
| *The tutor's role in the tutorial group* | | | | | | | | | |
| Students need my feedback to support their learning | - | - | 4.3 (1) | 52.2 (12) | 43.5 (10) | 4.39 | 0.58 | 3 | 5 |
| I tend to explain and teach the tutorial group | - | - | 4.3 (1) | 60.9 (14) | 34.8 (8) | 4.3 | 0.56 | 3 | 5 |
| My role as tutor is usually passive in the tutorial group | 26.1 (6) | 52.2 (12) | 8.7 (2) | 13.0 (3) | - | 2.09 | 0.95 | 1 | 4 |
| The students find it difficult to judge the relevance of literature found | - | 43.5 (10) | 13.0 (3) | 34.8 (8) | 8.7 (2) | 3.09 | 1.08 | 2 | 5 |
| *Relationship between theory and practice in PBL* | | | | | | | | | |
| PBL motivates me to continuously update my skills as a teacher | - | 4.3 (1) | - | 39.1 (9) | 56.5 (13) | 4.48 | 0.73 | 2 | 5 |
| PBL is based on true-life cases which creates involvement | - | 4.3 (1) | - | 52.2 (12) | 43.5 (10) | 4.35 | 0.71 | 2 | 5 |
| PBL creates a balance between theory and practice in education | - | 4.3 (1) | - | 47.8 (11) | 47.8 (11) | 4.39 | 0.72 | 2 | 5 |
| I was selected by my department | 4.3 (1) | 8.7 (2) | 21.7 (5) | 43.5 (10) | 21.7 (5) | 3.7 | 1.06 | 1 | 5 |
| *Assertions* | | | | | | | | | |
| The students' motivation level affects work in the tutorial group | - | - | - | 78.3 (18) | 21.7 (5) | 4.22 | 0.42 | 4 | 5 |
| PBL stimulates student learning | - | - | - | 52.2 (12) | 47.8 (11) | 4.48 | 0.51 | 4 | 5 |
| Tutorial groups help students to share experiences with each other | - | - | 4.3 (1) | 56.5 (13) | 39.1 (9) | 4.35 | 0.57 | 3 | 5 |
| I try to create a positive working atmosphere in the tutorial group | - | - | - | 34.8 (8) | 65.2 (15) | 4.65 | 0.49 | 4 | 5 |
| Group discussion facilitates problem processing | - | - | 4.3 (1) | 52.2 (12) | 43.5 (10) | 4.39 | 0.58 | 3 | 5 |
| I intervene and redirect discussion if it takes a wrong turn | 8.7 (2) | 4.3 (1) | - | 52.2 (12) | 34.8 (8) | 4 | 1.17 | 1 | 5 |
| Group meetings enrich student learning through communication | - | 4.3 (1) | 4.3 (1) | 43.5 (10) | 47.8 (11) | 4.35 | 0.78 | 2 | 5 |
| I support learning by helping students perform the learning tasks | - | - | - | 65.2 (15) | 34.8 (8) | 4.35 | 0.49 | 4 | 5 |
| I see to it that all students have their say in the group | - | 4.3 (1) | 13.0 (3) | 47.8 (11) | 34.8 (8) | 4.13 | 0.81 | 2 | 5 |
| Tutorial group size is right from a tutorial point of view | - | 8.7 (2) | 17.4 (4) | 47.8 (11) | 26.1 (6) | 3.91 | 0.9 | 2 | 5 |
| I help the students to achieve the objectives of the course | - | - | - | 52.2 (12) | 47.8 (11) | 4.48 | 0.51 | 4 | 5 |
| I am sensitive to the wishes of students when they need support | - | - | - | 34.8 (8) | 65.2 (15) | 4.65 | 0.49 | 4 | 5 |

<div align="center">

**Table 4.** *Cont.*

</div>

| | 1—Fully Disagree to 5—Fully Agree | | | | | | | | |
|---|---|---|---|---|---|---|---|---|---|
| **Questions** | **Frequency Distribution % (n)** | | | | | **Statistical Characteristics** | | | |
| **PBL as Pedagogical Method** | **1** | **2** | **3** | **4** | **5** | **M** | **SD** | **Min** | **Max** |
| | **Fully Disagree** | **Disagree** | **I Don't Know** | **Agree** | **Fully Agree** | | | | |
| *Items/Opinions* | | | | | | | | | |
| I often have problems with group dynamics in the tutorial group | 8.7 (2) | 56.5 (13) | 8.7 (2) | 17.4 (4) | 8.7 (2) | 2.61 | 1.16 | 1 | 5 |
| Work in the group has a test function and is stressful for students | 13.0 (3) | 69.6 (16) | 4.3 (1) | 13.0 (3) | - | 2.17 | 0.83 | 1 | 4 |
| Time for tutorial group work is too short | 4.3 (1) | 52.2 (12) | 8.7 (2) | 21.7 (5) | 13.0 (3) | 2.87 | 1.22 | 1 | 5 |
| I tend to explain and teach the group | - | 4.3 (1) | 17.4 (4) | 65.2 (15) | 13.0 (3) | 3.87 | 0.69 | 2 | 5 |
| *Assertions* | | | | | | | | | |
| I prefer PBL instead of classical teaching methods | - | 4.3 (1) | 8.7 (2) | 43.5 (10) | 43.5 (10) | 4.26 | 0.81 | 2 | 5 |
| There are a lot of advantages of PBL in compare with classical teaching | - | - | 4.3 (1) | 47.8 (11) | 47.8 (11) | 4.43 | 0.59 | 3 | 5 |
| I prefer to use PBL in my teaching in the future as well | - | - | - | 52.2 (12) | 47.8 (11) | 4.48 | 0.51 | 4 | 5 |
| I was confident in applying PBL approach | - | 8.7 (2) | 8.7 (2) | 56.5 (13) | 26.1 (6) | 4 | 0.85 | 2 | 5 |
| I was good prepared for teaching using the PBL | - | 4.3 (1) | 17.4 (4) | 56.5 (13) | 21.7 (5) | 3.96 | 0.77 | 2 | 5 |
| I need some additional trainings for better applying the PBL approach | 4.3 (1) | - | 8.7 (2) | 56.5 (13) | 30.4 (7) | 4.09 | 0.9 | 1 | 5 |
| It was easy to encourage students to apply PBL | 4.3 (1) | 13.0 (3) | 4.3 (1) | 69.6 (16) | 8.7 (2) | 3.65 | 0.98 | 1 | 5 |
| I need to familiarize with the specific materials for teaching using PBL | - | 8.7 (2) | 8.7 (2) | 47.8 (11) | 34.8 (8) | 4.09 | 0.9 | 2 | 5 |
| *Gender* | NA | NA | NA | NA | NA | 1.61 | 0.49 | 1 | 2 |
| *Age* | NA | NA | NA | NA | NA | 35.52 | 6.95 | 27 | 52 |
| *Years of experience in teaching* | NA | NA | NA | NA | NA | 8.87 | 7.36 | 1 | 27 |

Reliability statistics are shown in Table 5. As is clear from this table, Cronbach's Alpha was above 0.8 points. Thus, the questionnaire had a good reliability.

**Table 5.** Reliability statistics: Cronbach's alpha result of tutors' survey.

| Reliability Statistics | | |
|---|---|---|
| **Cronbach's Alpha** | **Cronbach's Alpha Based on Standardized Items** | **$n$ of Items** |
| 0.77 | 0.809 | 58 |

The factor analysis for the tutors' survey suggested five factors, but as there were very few questions belonging to each factor, we decided to restrict it to only two factors. The results of the factor analysis of tutors' survey are shown below. Table 6 describes the factor analysis of tutors' main survey. Total variance and component transformation matrix are also shown below (Tables A1 and A2).

**Table 6.** Factor analysis of tutors' survey.

| Question # in Questionnaire | Question | Factor 1 | Factor 2 | Factors | |
|---|---|---|---|---|---|
| | | | | **Factor 1** | **Factor 2** |
| Q_8 | PBL helps the student acquire relevant knowledge for their profession | **0.367** | −0.099 | ✓ | |
| Q_9 | PBL contributes to the independence of students | **0.62** | 0.499 | ✓ | |
| Q_10 | Group tutorials help students to evaluate their own knowledge | **0.432** | 0.056 | ✓ | |
| Q_11 | Group tutorials enrich student learning (communication and reflection) | **0.536** | 0.158 | ✓ | |
| Q_12 | Group tutorials help students share experiences with each other | **0.613** | 0.024 | ✓ | |
| Q_13 | In group tutorials the students have time to sort out issues that are hard to understand | **0.682** | −0.116 | ✓ | |
| Q_14 | Group discussions help problem processing | 0.437 | **0.575** | ✓ | ✓ |
| Q_15 | Work in tutorial group helps students to reach an optimal depth of knowledge | **0.659** | 0.065 | ✓ | |
| Q_16 | In my opinion, PBL is a great tool for student learning | **0.455** | 0.043 | ✓ | |
| Q_17 | I support student learning by helping them to achieve the learning goals | 0.012 | **0.196** | | ✓ |
| Q_18 | Supervising problem processing I help the students to fulfill the aims of the course | 0.331 | **−0.421** | | ✓ |
| Q_19 | I function as a resource person in the group | 0.08 | **0.24** | | ✓ |
| Q_20 | I participate in creating a positive work environment for the group | 0.072 | **0.579** | | ✓ |
| Q_21 | I encourage student learning by stimulating questions | −0.164 | **0.331** | | ✓ |
| Q_22 | I stress the importance of constant student reflection | −0.068 | **0.228** | | ✓ |
| Q_23 | I see to it that all students in the group have their say | **0.28** | 0.16 | ✓ | |
| Q_24 | I am sensitive to the wishes of the students regarding their need for support | **0.305** | 0.105 | ✓ | |
| Q_25 | I am interested in being a tutor | 0.215 | 0.225 | | |
| Q_26 | I have relevant teaching qualifications in PBL | −0.395 | **0.412** | | ✓ |
| Q_27 | It is difficult for students to know if they have learned enough | 0.008 | **0.426** | | ✓ |
| Q_28 | Discussions in the tutorial group are slow-moving | −0.513 | 0.543 | | |
| Q_29 | Work in the tutorial group has a test function and is stressful for students | **0.278** | 0.193 | ✓ | |
| Q_30 | Time for discussion in the tutorial group is too short | 0.143 | 0.142 | | |

**Table 6.** *Cont.*

| Question # in Questionnaire | Question | Factor 1 | Factor 2 | Factors | |
|---|---|---|---|---|---|
| | | | | Factor 1 | Factor 2 |
| Q_31 | The group size is just right from a tutorial point of view | 0.406 | **0.557** | | ✓ |
| Q_32 | Discussion in the tutorial group creates uncertainty among students | **0.278** | −0.126 | ✓ | |
| Q_33 | PBL evokes feelings of inadequacy in students | −0.107 | **0.647** | | ✓ |
| Q_34 | Students need my feedback to support their learning | −0.295 | **0.525** | | ✓ |
| Q_35 | I tend to explain and teach the tutorial group | −0.042 | 0.662 | | ✓ |
| Q_36 | My role as tutor is usually passive in the tutorial group | −0.279 | 0.255 | | |
| Q_37 | The students find it difficult to judge the relevance of literature found | −0.008 | **0.185** | | ✓ |
| Q_38 | PBL motivates me to continuously update my skills as a teacher | **0.578** | −0.132 | ✓ | |
| Q_39 | PBL is based on true-life cases which creates involvement | **0.61** | −0.236 | ✓ | |
| Q_40 | PBL creates a balance between theory and practice in education | **0.472** | −0.072 | ✓ | |
| Q_41 | I was selected by my department | **0.271** | −0.171 | ✓ | |
| Q_42 | Group works are very important in deep learning | 0.271 | **0.682** | | ✓ |
| Extraction method: principal component analysis. Rotation method: Varimax with Kaiser Normalization. | | | | | |
| a. Rotation converged in 3 iterations. | | | | | |

Factor 1 includes questions Q8–Q13, Q15, Q16, Q23, Q24, Q29, Q32, and Q38–Q41, while factor 2 includes Q14, Q17-Q22, Q26, Q27, Q31, Q33, Q34, Q35, Q37, and Q42. Factor 1 refers to tutors' satisfaction level with PBL, which is high. They believe that PBL can develop students' ability in group/team working. Factor 2 refers to tutors' perception of PBL as a factor to improve their skills during the educational process. This refers to learning outcomes and tutors' skills improvement during the educational process and supervising.

In the open question (Supplementary Materials S#4 question 16), tutors mentioned some difficulties and problems such as the absence of necessary equipment in the auditorium, lack of connection to worldwide well-known virtual platforms, and lack of video and presentations from previous tutors. Except for some technical issues, they also mentioned barriers such as problems with student engagement and motivation, as well as lack of student experience with PBL. At the same time, a few tutors mentioned the absence of their own experience with PBL and that it does not enable them to be sufficiently effective and professional. They felt a need to visit other universities where PBL is applied. Besides these, tutors noticed the additional challenges related to the pandemic.

## 4. Discussion

It is evident from the previous literature that the implementation of the Bologna Process was not successful as most efforts to modernize the curriculum and teaching methods faced huge obstacles and resistance because of educational traditions [17–19]. Classical traditional approaches in education have become rooted in the system, which made reforms difficult. The reason is that mainly fundamental/theoretical subjects and specialties are useful in a plan-oriented approach but are not in a market-oriented economy. To the best of our knowledge, few articles have addressed this topic [4]. An ageing academic staff, a non-competitive culture, and the unattractiveness of being employed in higher education for younger generations, among others, explain this reluctance to change. According to a study in Kyrgyzstan, whose historical and mental background is quite similar the lecture-centered learning, is still prevalent, and the 'jump' to a new system with a more modern approach is not easy for academic staff [20], especially if we take into account that the majority of staff including administrative staff also received their education from kindergarten up to university in the Soviet tradition. Walz et al. [20]

describes numerous challenges in the post-Soviet country Kyrgyzstan and the difficulties with the modernization of higher education. They conclude that "Taking the first steps to explore and implement open pedagogy may be a challenge for instructors, particularly with its strong emphasis on student agency, active, engaged student learning, the instructor as a 'coach,' and interaction with curation and creating in the 'real world' rather than assignments that mainly the teacher sees".

During our PBL pilot semester, we also encountered a number of challenges. There were auditoriums with broken or absent necessary equipment, lack of internet connection or the inability to use well-known virtual platforms, and a lack of a video database with podcasts or multimedia files from previous tutors. Aside from these technical issues, there were other barriers such as problems with student engagement and motivation. In particular, this was evident during the distant learning phase because of the pandemic. At the same time, there was not enough knowledge and experience among students with PBL. There was an issue with a few tutors who clearly were not confident with PBL and who claimed that it did not enable them to be sufficiently effective and professional. They expressed a need to visit other universities where PBL is applied. Besides these issues, some tutors mentioned additional difficulties related to the unexpected fully distance mode because of COVID-19.

Reform of the curriculum in Azerbaijan is necessary as the current way of teaching does not properly prepare students for the needs of the labor market. Practical and applied subjects and specialties do not exist and are not developed. Thus, the curriculum is fully theoretical, and practical knowledge was very limited. Necessary competencies, skills, and learning outcomes are not part of the higher education system. Difficult and deep theoretical knowledge is the major pillar of high education [5]. Finally, the inheritance of the culture of socialism plays a role as well. As a result, Azerbaijan has a very traditional culture of teaching in higher education, one that is mainly oriented around the lecturer and focused on theoretical knowledge. However, many initiatives for renewal have been initiated in Azerbaijan.

The PBL pilot intervention that we conducted is one of these renewals. We found that according to students and tutors' opinions, this helped students develop necessary skills. Moreover, such initiatives trigger the process of modernization of higher education. At the same time, the application of PBL in engineering-based specialties helps both tutors and students to be more focused on real cases and the challenges of industry. Tutors' tips and creativity helps their colleagues from the department also. We experienced this during different sessions as other tutors participate at events through their colleagues. Tutors asked permission for participation of their colleagues as they were interested as well. The above-mentioned results show that PBL has the potential to become accepted in Azerbaijan. This fact is important not only for this country, but for the region itself and other countries with traditional teaching methods. However, despite the problems and similarities in the outdated teaching methods in post-Soviet countries, students and staff [17,21] from countries with a similar background and history with a similar culture of curriculum seem to agree on the fruitfulness of the modernization of higher education. The authors mention the positive impact on learning effectiveness and motivation if students feel themselves leading in the process of their own education [17]. Moreover, this directly influences their self-development and ability to work in tutorial groups [21]. This approach supports students in increasing necessary competences and skills for the right 'track' in their future career development [18]. Another study in the Russian Federation also argued that the PBL method triggers students to learn new things more quickly; helps students to adapt to new real cases more easily; and helps them to work independently and to demonstrate skills in computer technologies and communication, teamwork, critical-thinking, creativity, etc. [19].

According to government programs of the Azerbaijan Republic, thousands of young people have the opportunity to study abroad. We expect that after their study and coming back to Azerbaijan and that their active involvements in the higher education process both as tutors or as administrative staff members will have its advantages. Curriculum

development is ongoing in many universities of the country. Erasmus+ projects provide a positive impact as well. Government and business are also interested in internship programs and volunteering work. Furthermore, a young, modern, and open-minded generation outside academia will influence the academic community, as well as university top management. The same holds for new generations within the university.

Moreover, the market economy has a positive impact upon competition. Universities are interested in recruiting more students and to be attractive for the best students. Success in employability and career of graduates' influence on high interest from the enrollees' side directly. Thus, universities are motivated to modernize [5]. At the same time, applying PBL inadequately is full of risks [22,23]. This was shown by our findings as well. The unskillful application of PBL might confuse tutors and disrupt the learning process of students. There are studies that describe such negative outcomes [23] and show a low level of satisfaction with PBL because of a lack of preparation and a need for adequate regulation of didactical innovations.

This we recognize in our PBL pilot as well. Because of the lack of previous experience with PBL during education (primary and high schools are also traditional), there is a problem in understanding the basic pillars of PBL such as interactive learning grounded on evidence-based research. The focus on traditional education with its repetition of tutors' lecture and learning by heart theoretical information is questioned. Nevertheless, the positive thoughts and attitudes of both of students and tutors suggest that innovation in education should be continued and might over time become more fruitful.

## 5. Limitations

This study used a cross-sectional design. This implies some limitations; for example, it does not warrant causal inferences. Moreover, this study did not evaluate the effects of the application of PBL in AzUAC. The evaluation of the effectiveness of PBL needs additional study and analysis of learning outcomes. We leave it to further research to compare the outcomes of traditional teaching methods with outcomes of the PBL approach. This can be done by using experimental design in analyzing the learning outcomes.

## 6. Conclusions

During the one-semester-long intervention period, tutors were using PBL instead of the traditional way of teaching. We were interested in the satisfaction with the use of PBL. Tutors had many challenges to overcome, including a lack of experience in working with PBL, and this was exacerbated by the COVID-19 pandemic and the switch to online learning. In total, 424 (401 students and 23 tutors) people successfully finished the PBL semester and participated in the surveys. The majority of the students, 395 persons, expressed satisfaction with one of the main pillars of PBL, the tutorial groups. As for subject content, more than 44% of students thought that it was difficult or even too difficult, while 13% thought that it was too easy. Thus, around 50% of students were not quite satisfied with the subject content.

This is a positive finding, taking into account the fact that there was no practical experience with PBL before from both sides of education and that the pilot tutors were front-runners among the staff community. It is also clear that more than 82% of the students thought that the PBL sessions should be kept as part of this module, while 4.7% of them disagreed with this. This is in accordance with tutors, among who 91.3% agreed that PBL is a great tool for student learning.

**Supplementary Materials:** The following are available online at https://www.mdpi.com/article/10.3390/educsci11060288/s1.

**Author Contributions:** Conceptualization, U.S., W.G. and J.A.; methodology, W.G. and J.A.; formal analysis, U.S., W.G. and J.A.; investigation, U.S., W.G., J.A.; resources, U.S., W.G. and J.A.; data curation, U.S.; writing—original draft preparation, U.S., W.G., J.A.; writing—review and editing W.G. and J.A.; supervision, W.G. and J.A.; All authors have read and agreed to the published version of the manuscript.

**Funding:** This research received no external funding.

**Institutional Review Board Statement:** Not applicable.

**Informed Consent Statement:** Informed consent was obtained from all subjects involved in the study.

**Data Availability Statement:** Data available in a publicly accessible repository.

**Conflicts of Interest:** The authors declare no conflict of interest.

## Appendix A

| TASK NAME | START DATE | DUE DATE | DURATION |
|---|---|---|---|
| **Preparation stage** | 21.01.2020 | 31.01.2020 | 10 |
| **First stage (formalizing "PBL Group")** | 03.02.2020 | 14.02.2020 | 11 |
| 1 staff's survey | | | |
| Info day about PBL | | | |
| Motivation the staff | | | |
| Share experience | | | |
| Membership of "PBL group" | | | |
| **Second stage (Start "unusual" semester)** | 17.02.2020 | 28.02.2020 | 11 |
| Informing academic community and administrative staff concerning application of PBL | | | |
| 1 students' survey | | | |
| **Third stage (Managing the process of PBL installation)** | 17.02.2020 | 03.06.2020 | 107 |
| Team building process was prolonged till the end of semester | | | |
| Trainings/workshops & talks | | | |
| Motivating staff to improve skills for PBL teaching | | | |
| **Fourth stage (Finish "unusual" semester)** | 24.05.2020 | 28.07.2020 | 65 |
| 2 students survey | | | |
| 2 staff survey | | | |
| Feedbacks & discussions | | | |
| Data collection and sorting data | | | |
| Analytical work using spss tool | | | |
| Final discussions concerning received results | | | |

**Figure A1.** The scientific research design of PBL installation in AzUAC.

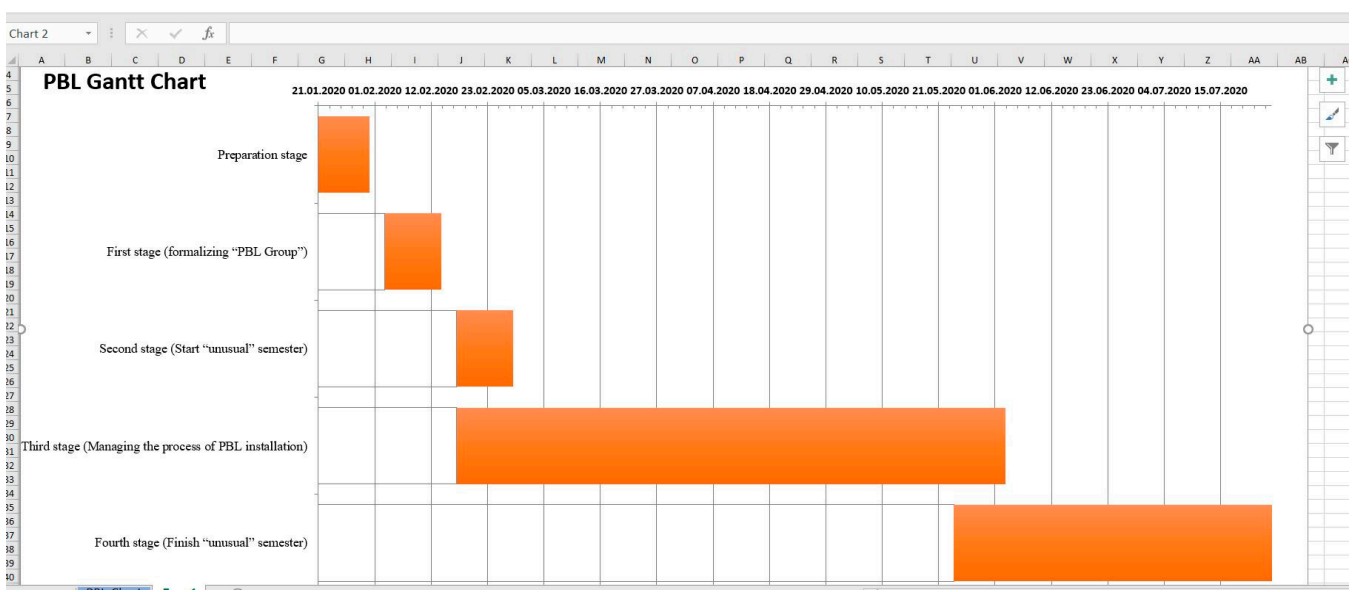

**Figure A2.** PBL Gantt chart.

**Table A1.** Component Transformation Matrix.

| Component | 1 | 2 |
|---|---|---|
| 1 | 0.716 | 0.698 |
| 2 | 0.698 | −0.716 |
| Extraction method: principal component analysis. | | |
| Rotation method: Varimax with Kaiser normalization. | | |
| Extraction method: principal component analysis. | | |

**Table A2.** Component Transformation Matrix.

| Component | 1 | 2 |
|---|---|---|
| 1 | 1000 | −0.003 |
| 2 | 0.003 | 1000 |
| Extraction method: principal component analysis. | | |
| Rotation method: Varimax with Kaiser normalization. | | |

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
