# Peer review of "Student and Tutor Satisfaction with Problem-Based Learning in Azerbaijan"

_education, doi:10.3390/educsci11060288_

Round 1

Reviewer 1 Report

The introduction describes very well the university context of Azerbaijan and its problems with the introduction of innovative teaching methodologies. Given the need to train students for the new social and labor demands, it is necessary to make changes in the teaching practices of higher education. However, a review of knowledge about the PBL is missing. It is necessary to justify the relevance of this methodology and the main research results that support its use in higher education.
On the other hand, it would be necessary to offer a brief description of the specific use of the PBL with the students (activities, tools, resources, organization, ...)
The results are presented clearly and rigorously. The data analysis technique is considered adequate to the objectives of the study. The instruments are well developed, although stronger theoretical support would be desirable. The results are of interest, especially in the context of the Azerbaijani universities. Its dissemination can promote educational change in universities. The discussion does not introduce many references to contrast the results. The authors limit themselves to citing the works of researchers from the close geographical and cultural context, but they could use other broader references. The study limitations stated by the authors are relevant.

Author Response

Dear Reviewer,

Thank you for the critical comments that you have provided on our paper. These comments have greatly helped us improve our work. Below we describe point-by-point how we have addressed your comments. Page numbers refer to the revised document. We also adjusted the manuscript to accommodate the' comments. The changes are highlighted by the “track checker” in the manuscript as well.

Comment

The introduction describes very well the university context of Azerbaijan and its problems with the introduction of innovative teaching methodologies. Given the need to train students for the new social and labor demands, it is necessary to make changes in the teaching practices of higher education. However, a review of knowledge about the PBL is missing. It is necessary to justify the relevance of this methodology and the main research results that support its use in higher education.
On the other hand, it would be necessary to offer a brief description of the specific use of the PBL with the students (activities, tools, resources, organization, ...)

Reply

We add some more information about the PBL approach in the introduction part of the paper  to better justify the relevance of this approach and its usage in higher education. We do this by reviewing some of the previous literature. These studies have shown that PBL is widely applied in higher education (see ref. of the paper 1-4, 8-23). Besides, we added information about the main tips and cases, which were eligible and cited to the paper, which were not included in the previous draft. Thus, we added this in the introduction section (page 3).

Comment
The results are presented clearly and rigorously. The data analysis technique is considered adequate to the objectives of the study. The instruments are well developed, although stronger theoretical support would be desirable. The results are of interest, especially in the context of the Azerbaijani universities. Its dissemination can promote educational change in universities. The discussion does not introduce many references to contrast the results. The authors limit themselves to citing the works of researchers from the close geographical and cultural context, but they could use other broader references. The study limitations stated by the authors are relevant.

Reply:

Thank you very much for your comments and contribution to the article. We have added a brief description of previous studies on student satisfaction with  PBL on page 1 of the introduction section to justify the relevance of this approach in higher education.

On your suggestion we added more references throughout the paper as well.

In the discussion section we also made more frequent reference to other studies on this topic (pages 14 and 15).

Reviewer 2 Report

From the point of view of structure, the article is well done.
In terms of content it is very weak.
There is a difference between problem-based learning and the application of a satisfaction questionnaire.
The method does not mainly explain how problem-based learning has been applied. In my opinion, this is the main subject of the didactic activity. The satisfaction part is less important.
The article presents a statistic based on the questionnaire.
The statistical calculation, well done, is not likely to present the innovative elements of the subject regarding problem-based learning.
No assessments are made on the same subject in other studies in the field of education in the same specialty or in other specialties.
Bibliographic references are few.

The specialized literature offers many many sources that have as subject the didactic strategies like problem-based learning in different fields of study. This could better support the fundamental part of article.

Author Response

Dear Reviewer

Thank you for the critical comments that you have provided on our paper. These comments have greatly helped us improve our work. Below we describe point-by-point how we have addressed your comments. Page numbers refer to the revised document. We also adjusted the manuscript to accommodate the' comments. The changes are highlighted in the manuscript as well.

Comment:

From the point of view of structure, the article is well done.
In terms of content it is very weak.
There is a difference between problem-based learning and the application of a satisfaction questionnaire.
The method does not mainly explain how problem-based learning has been applied. In my opinion, this is the main subject of the didactic activity. The satisfaction part is less important.
The article presents a statistic based on the questionnaire.
The statistical calculation, well done, is not likely to present the innovative elements of the subject regarding problem-based learning.

Reply:

We began to apply PBL first and after that conducted the satisfaction survey among academic staff members and students. We continue our research now. Currently there is the third semester where PBL is applied. We are analyzing the learning outcomes of PBL. We hope to publish these results also. As to the PBL methods, we continu working on this issue as well by various workshops, boot camps that are organized periodically both for students and tutors. We try to use different platforms for developing this. To account for this, we added information about the main tips and cases. We have also expanded on our reference list. Thus, in the introduction section  (page 3) we now mention seminars and other tutorial events.

The main purpose of this study is to measure the satisfaction with the PBL approach among students and tutors. We do not do evaluate PBL but we look at the satisfaction. When the students and staff are satisfied with PBL, it is expected to,facilitate the implementation of it. To account for this we rewrote the method section from the first paragraph to make the purpose of the research clearer (pages 3-4).

Comment:
No assessments are made on the same subject in other studies in the field of education in the same specialty or in other specialties.

Bibliographic references are few.

Reply:

We have adjusted the manuscript and now provide more context by providing an overview of the results of other studies in the introduction, conclusion and discussion sections (pages 1, 2, 13,14,15,16,17). We have also added more references in the text.

Comment:

The specialized literature offers many many sources that have as subject the didactic strategies like problem-based learning in different fields of study. This could better support the fundamental part of article.

Reply:

We now discuss the findings from previous studies more extensively and have included more references in the paper. This has been done on pages (pages 1, 2, 13,14,15,16,17).

Reviewer 3 Report

The present manuscript is valuable research focused on the didactic framework of problem-based learning. It offers results that may interest a large part of the scientific community in higher education's pedagogical field. However, we believe the article needs several improvements, which are described below:

  1. It would be desirable that the abstract be reworked without establishing a division between the different parts of the manuscript (objective, method, results...), which is more typical of experimental and health sciences. Naturally, all these sections should be showed. Still, it is unnecessary to divide them using a colon.

  1. It would be convenient to use abbreviations. Problem-Based Learning" cannot be used continuously in the abstract. The correct thing to do is to use the acronym the first time "Problem-Based Learning (PBL)" is mentioned and only indicates this abbreviation.

  1. The fundamental problem of this manuscript is that it lacks the slightest review of the scientific literature and the use of sources. An exhaustive analysis of previous works related to the problem is necessary. Furthermore, not only should the state of the question be established in the "Introduction" section, but also the "Discussion" section, it is necessary to establish precisely what is innovative in terms of work, what it offers about other previous contributions, compare it with what has been done in other countries, etc.

  1. To this end, we recommend that there should be a specific "Discussion" section before the conclusions, where the achievements of the research carried out should be precisely established.

  1. References do not use the journal's citation systems. It is necessary to modify the references according to MPDI standards and make them and those incorporated, which should be many because of the shortcomings of the review of scientific literature.

Author Response

Dear reviewer

Thank you for the critical comments that you have provided on our paper. These comments have greatly helped us improve our work. Below we describe point-by-point how we have addressed your comments. Page numbers refer to the revised document. We also adjusted the manuscript to accommodate the' comments. The changes are highlighted in the manuscript as well.

Comment

The present manuscript is valuable research focused on the didactic framework of problem-based learning. It offers results that may interest a large part of the scientific community in higher education's pedagogical field. However, we believe the article needs several improvements, which are described below:

Reply

Thank you for the words of appreciation of our work.

Comment

  1. It would be desirable that different parts of the manuscript (objective, method, results...), which is more typical of experimental and health sciences. Naturally, all these sections should be showed. Still, it is unnecessary to divide them using a colon.

 Reply

We have adjusted the abstract  (page 1). We reworked the abstract without establishing a division between the parts of the paper.

Comment

  1. It would be convenient to use abbreviations. Problem-Based Learning" cannot be used continuously in the abstract. The correct thing to do is to use the acronym the first time "Problem-Based Learning (PBL)" is mentioned and only indicates this abbreviation.

 Reply:

We changed the manuscript accordingly. Thus, we used the PBL acronym throughout the text after writing it in full at the first use of the term (page 1), so instead of writing “Problem-Based Learning” PBL is used everywhere now.  

Comment

  1. The fundamental problem of this manuscript is that it lacks the slightest review of the scientific literature and the use of sources. An exhaustive analysis of previous works related to the problem is necessary. Furthermore, not only should the state of the question be established in the "Introduction" section, but also the "Discussion" section, it is necessary to establish precisely what is innovative in terms of work, what it offers about other previous contributions, compare it with what has been done in other countries, etc.

 Reply

We have corrected this and added more text discussing previous work on PBL now. For this we have rewritten the introduction, discussion and conclusion parts (pages 1,2,3,14,15,16).

We have also included more references in the paper.

Comment

  1. To this end, we recommend that there should be a specific "Discussion" section before the conclusions, where the achievements of the research carried out should be precisely established

Reply

We have divided the disscussion and conclusion sections. Thus, we added a discussion section in which we disscussed the main findings of our study (pages 14,15) . At the same time we revised the conclussion (pages 15,16) section as well.

Comment

  1. References do not use the journal's citation systems. It is necessary to modify the references according to MPDI standards and make them and those incorporated, which should be many because of the shortcomings of the review of scientific literature

Reply

We have adjusted the reference  according to MDPI standards ( the reference list pages 16, 17).

Round 2

Reviewer 3 Report

The authors have made the requested and recommended changes. There are still some typos (mostly formatting), but I understand that they will be corrected in proofreading if the article is finally accepted.